# Intracerebral Injection of Extracellular Vesicles from Mesenchymal Stem Cells Exerts Reduced Aβ Plaque Burden in Early Stages of a Preclinical Model of Alzheimer’s Disease

**DOI:** 10.3390/cells8091059

**Published:** 2019-09-10

**Authors:** Chiara A. Elia, Matteo Tamborini, Marco Rasile, Genni Desiato, Sara Marchetti, Paolo Swuec, Sonia Mazzitelli, Francesca Clemente, Achille Anselmo, Michela Matteoli, Maria Luisa Malosio, Silvia Coco

**Affiliations:** 1Laboratory of Pharmacology and Brain Pathology, Neuro Center, Humanitas Clinical and Research Center -IRCCS- Via Manzoni 56, 20089 Rozzano, Italy; 2CNR, Institute of Neuroscience, Via Vanvitelli 32, 20129 Milano, Italy; 3Humanitas University, Department of Biomedical Sciences, Via Rita Levi Montalcini 4, 20090 Pieve Emanuele, Italy; 4CNR, Institute of Neuroscience, Via Moruzzi 1, 56125 Pisa, Italy; 5School of Medicine and Surgery and Milan Center for Neuroscience (NeuroMI), University of Milano-Bicocca, Via Cadore, 48, 20900 Monza, Italy; 6Cryo-Electron Microscopy Lab, Department of Biosciences, University of Milan, Via Celoria 26, 20133 Milano, Italy; 7Centro di Ricerca Pediatrica Romeo ed Enrica Invernizzi, University of Milano, Via Celoria 26, 20133 Milano, Italy; 8Flow Cytometry Core Facility, Humanitas Clinical and Research Center, Via Manzoni 56, 20089 Rozzano, Italy

**Keywords:** bone marrow mesenchymal stem cells, extracellular vesicles, Alzheimer’s disease, APPswe/PS1dE9 AD mice, Neprilysin, dystrophic neuritis, SMI, Aβ plaques

## Abstract

Bone marrow Mesenchymal Stem Cells (BM-MSCs), due to their strong protective and anti-inflammatory abilities, have been widely investigated in the context of several diseases for their possible therapeutic role, based on the release of a highly proactive secretome composed of soluble factors and Extracellular Vesicles (EVs). BM-MSC-EVs, in particular, convey many of the beneficial features of parental cells, including direct and indirect β-amyloid degrading-activities, immunoregulatory and neurotrophic abilities. Therefore, EVs represent an extremely attractive tool for therapeutic purposes in neurodegenerative diseases, including Alzheimer’s disease (AD). We examined the therapeutic potential of BM-MSC-EVs injected intracerebrally into the neocortex of APPswe/PS1dE9 AD mice at 3 and 5 months of age, a time window in which the cognitive behavioral phenotype is not yet detectable or has just started to appear. We demonstrate that BM-MSC-EVs are effective at reducing the Aβ plaque burden and the amount of dystrophic neurites in both the cortex and hippocampus. The presence of Neprilysin on BM-MSC-EVs, opens the possibility of a direct β-amyloid degrading action. Our results indicate a potential role for BM-MSC-EVs already in the early stages of AD, suggesting the possibility of intervening before overt clinical manifestations.

## 1. Introduction

Alzheimer’s disease (AD), the leading cause of dementia, has recently been attracting a lot of attention from the scientific community, since millions of people are affected by this incurable pathology; furthermore, the number of patients is destined to increase in the coming decades [1]. The limited knowledge of the etiology of Alzheimer’s has rendered in vain numerous attempts hitherto pursued to find a resolutive treatment that is not simply limited to the alleviation of symptoms. Therefore, a worldwide effort is underway to discover the mechanisms responsible for the disease onset and progression and to find an efficacious therapy, developing either novel treatments [2,3,4] or preventive strategies [5]. 

Cell therapy is becoming a new reality for many diseases. Due to the plasticity and multifaceted features of stem cells, recent studies have also focused on their possible exploitation in AD [6,7,8]. Multipotent bone marrow mesenchymal stem cells (BM-MSCs) represent a heterogeneous subset of stromal cells. They can be isolated from bone marrow or many other adult tissues, including periosteum, trabecular bone, adipose tissue, synovium, pancreas, placenta and cord blood [9]. They typically give rise to cells of diverse lineages, including adipocytes, chondrocytes and osteocytes. In recent years, BM-MSCs have been shown to be endowed with immunoregulatory abilities [10]. This property makes them suitable as possible therapeutic tools for AD [11], a disease characterized by a large inflammatory component mediated by microglia activation [2]. Interestingly, transplantation of human adipose tissue-derived MSCs (ADSC) into the brain has been shown to reduce Aβ deposition and to restore microglial function in transgenic APPswe/PS1dE9 (APP/PS1) mice, a preclinical model widely used for the study of AD [12]. Moreover, BM-MSCs alleviated memory deficits in AD mice by modulating immune responses [13]. 

More recently the discovery of Extracellular Vesicles (EVs) originating from MSCs, retaining most of the properties of the cells of origin, is leading scientists to develop cell-free therapies to limit the potential side effects associated with the use of stem cells [14], including induction of vascular obstruction [15], lung retention after transplantation (resulting in a reduction in the population of cells that reach the target site) [16], the production of allo-antibodies was observed following repeated administration of MSCs [17], as well as a controversial protumorigenic effect [18]. In in vitro experiments, exosomes from human ADSC significantly decreased both secreted and intracellular Aβ levels in N2a cells engineered to overexpress human APP, by virtue of the proteolytic activity of neutral endopeptidase Neprilysin (NEP), the dominant Aβ peptide-degrading enzyme in the brain, towards Aβ_1-42_ peptide [19]. Human ADSC derived EVs were thus envisaged to represent a therapeutic tool based on their Aβ-degrading ability [19]. Another therapeutic perspective put forward is a possible role of EVs in Aβ_1-42_ scavenging, thanks to the ability of EVs glycosphingolipids to bind to Aβ and to convey it to the microglia for phagocytosis [20]. In addition, MSC-EVs have been described to exert an antiapoptotic, neuroprotective role [21,22] and to promote neurite outgrowth and axon regeneration of injured neurons [23,24]. Therefore, MSC-EVs, similarly to parental cells, are endowed with anti-inflammatory, amyloid-β degrading and neurotrophic activities that could stimulate neighboring parenchymal cells to start repairing damaged tissues. These properties have been considered very interesting to test in in vivo models of AD (for a recent review, see [11]). Altogether, this evidence suggests that MSC-derived EVs might play a therapeutic effect in AD. 

Despite the massive investments in AD drugs, the disappointing failure of several clinical trials conducted in recent years is forcing the neuroscience community to orient itself towards initiating the treatments at earlier stages of the pathology. Indeed, the pathophysiological process of AD is known to begin decades before diagnosis, with amyloid buildup occurring when only subtle clinical symptoms, if any at all, are evident [25]. 

In the present study, we hypothesized that the early therapeutic exploitation of MSC-EVs could be efficacious in addressing some of the disease features of AD, which may possibly slow, or even prevent, manifestation of the pathological signs. For this purpose, bone marrow mouse MSC-EVs were injected into the neocortex of APP/PS1 mice at two different time points: 5 months, when amyloid plaques were definitely present, and 3 months, when they were instead just starting to appear. Our results indicate that an early intervention with BM-MSC-EVs reduces pathological signatures of AD, thus suggesting that MSC-EVs could be regarded as a potentially effective treatment for the disease.

## 2. Materials and Methods

### 2.1. Culture of Purified Murine Bone Marrow Mesenchymal Stem Cells

Primary Bone Marrow Mesenchymal Stem Cells (BM-MSCs) were prepared from 4–12 week-old C57BL/6 mice by flushing femur and tibia bones cavities using plain culture medium, removing red blood cells by lysis with 0.84% (*w*/*v*) NH_4_Cl solution for 5 min at RT, then filtering through a 70 µm filter mesh and seeding them in tissue culture flasks in αMEM supplemented with 20% (*v*/*v*) FBS (HyClone, Cat.N. SH30070.03, GE Healthcare Bio-Sciences, Pittsburgh, PA, USA) in a 5% CO_2_ incubator. Nonadherent cells were removed 48 hours later by changing the medium. Cells were passaged at sub-confluency (~80%) with a split ratio of 1:3, or cryopreserved and stored at −80 °C in 10% DMSO in FBS. BM-MSCs were used from passages 9 to 14 (P9–P14). 

### 2.2. BM-MSC Osteogenic and Adipogenic Lineage Differentiation Assay

Differentiation of murine BM-MSCs towards the osteogenic and adipogenic lineages was performed in 6-well plates (5 × 10^3^ cells/well) with osteogenic differentiation medium (αMEM with 5% FBS, dexamethasone 10 nM, ascorbic acid 0.3 M, β-glycerophosphate 10 mM), adipogenic differentiation medium (αMEM with 5% FBS, dexamethasone 10 nM and insulin 0.5 μg/mL) or control medium (αMEM with 5% FBS). After 30 days, osteogenic and adipogenic differentiation were revealed by Alizarin Red and Red Oil O staining, respectively. 

### 2.3. Senescence Associated β-Galactosidase (SA-β-Gal) Assay

For SA-β-Galactosidase staining cells were washed in PBS and fixed in 2% (*w*/*v*) formaldehyde, 0.2% (*w*/*v*) glutaraldehyde for 5 min at RT and after 1 wash in PBS were incubated O/N in freshly prepared staining solution at pH 6.0 containing 1mg/mL Xgal (5-bromo-4-chloro-3-indolyl-β-galactopyranoside, Sigma-Aldrich^®^, Merck KGaA, Darmstadt, Germany) in 8 mM citric acid/sodium phosphate buffer, 150 mM NaCl, 2 mM MgCl_2_, 5 mM potassium ferrocyanide and 5 mM potassium ferricyanide. Cell nuclei were counterstained with Hoechst 33342 and visualized with a 20× objective on a wide field system (Olympus Cell^R system with an IX81 inverted microscope, Olympus, Hamburg, Germany) equipped with an MT20 illumination device with fluorescent filters (Ex: 361 nm; Em: 486 nm) and Differential Interference Contrast (DIC). For each condition, a minimum of 6 fields (containing on average of 50 cells) were acquired (Olympus Xcellence 1.2 RealTime controller software), and the percentage of β-Gal positive cells on the total number of cells was calculated after manually counting with Fiji ImageJ software the β-Gal positive cells that appeared with a black cytoplasm (DIC channel) and the Hoechst positive nuclei (Hoechst channel). The percentage of β-Gal positive cells on the total number of cells was plotted and subjected to statistical analysis with Graphpad Prism v.7.0 software.

### 2.4. Flow Cytometry Profiling of BM-MSCs 

Cells were stained using the appropriate saturating concentrations of the following conjugated antibodies: FITC-conjugated mouse hematopoietic lineage Cocktail (Lin, eBioscience™, San Diego, CA, USA), rat anti-mouse CD31 Brilliant-Violet™510-conjugated, rat anti-mouse Ly 6A/E (Sca1) phycoerythrin-conjugated, hamster anti-rat/mouse CD49a Alexa-Fluor^®^647-conjugated (BD Biosciences, San Jose, CA, USA), rat anti-mouse CD9 FITC-conjugated, rat anti-mouse/human CD44 Alexa-Fluor^®^647-conjugated, Armenian hamster anti-mouse/rat CD29 Pacific Blue™ conjugated, rat anti-mouse CD73 Alexa-Fluor^®^647-conjugated, rat anti-mouse CD105 PE/Cy7™-conjugated, rat anti-mouse CD117 (c-kit) PerCP™/Cy5.5-conjugated (BioLegend^®^, San Diego, CA, USA). Following surface staining, cells were fixed using 2% (*w*/*v*) paraformaldehyde (PFA) in PBS for 20 min on ice. An LSR Fortessa analyzer (BD Biosciences, San Jose, CA, USA), equipped with 4 lasers and able to discriminate up to 18 fluorophores, was used for sample acquisition. Instrument performance was checked daily using CS&T Beads (BD Biosciences, San Jose, CA, USA) and SPHERO Rainbow beads (Spherotech, Lake Forest, IL, USA). Data acquisition and analysis were performed with FACSDiva v.6.2 (BD Biosciences, San Jose, CA, USA) and Flow-Jo v.9.7 (Tree Star Inc., Ashland, OR, USA), respectively.

### 2.5. Isolation of BM-MSC-Derived EVs 

EVs were obtained from BM-MSC supernatants by a protocol adapted from [26], whereby a monolayer of approx. 3 × 10^6^ murine BM-MSCs, seeded in tissue culture flasks, was washed three times with PBS and incubated in serum-free αMEM for 3 h to induce EV release. The culture medium was then collected and centrifuged at 4 °C, first at 300 g for 10 min to discard dead cells, then at 1000× *g* for 20 min to eliminate cellular debris, and finally for 90 min at 110,000 g to pellet EVs composed of Exosomes and Microvesicles. For in vivo studies following removal of the centrifugation supernatant, EVs were rinsed in PBS and ultracentrifuged again before use. EV-enriched pellets were resuspended in PBS. For biochemical studies, EVs were solubilized in lysis buffer (10 mM HEPES-NaOH pH 7.5; 2 mM EDTA pH 8.0; 1% (*w*/*v*) SDS; supplemented with protease and phosphatase inhibitors) and protein content was quantified by micro-BCA assay (Thermo Fisher Scientific, Waltham, MA, USA).

### 2.6. BM-MSC-EV Characterization with Nanoparticle Tracking Analysis (NTA)

NTA measurements were performed with NanoSight NS300 (Malvern Panalytical, Malvern, UK) for determination of the size and concentration of isolated EV samples. Diluted samples (1:6 in 500 µL of PBS) were injected into the sample chamber with sterile syringes (BD Plastipak Insulin syringe, Franklin Lakes, NJ, USA) until the liquid reached the tip of the nozzle. All measurements were performed in dynamic mode, with syringe pump infusion rate 30, at room temperature and with the same viscosity value. EV samples and the EV-depleted media were analyzed right after isolation with manual shutter and gain adjustments. NTA 3.3 Dev Build 3.3.301 software was used for recordings, measurements and data output collection. Five dynamic measurements of each sample were performed (60 s each). Scripts were run via SOP-type procedures with default options for standard measurements. For each polydisperse sample, the average FTLA (particles/frame against size distribution) was considered. Raw particles size and concentration data were plotted and subjected to statistical analysis with GraphPad Prism 6.0^®^ or 7.0^®^ software. 

### 2.7. BM-MSC EV Profiling by Flow Cytometry 

To limit background noise from dust and salt crystals, 0.22-μm-filtered sheath fluid was used for sample acquisition. The FSC and SSC gate for identifying EVs was determined by using dimensional beads (size range: 0.1–0.9 µm, Biocytex, Marseille, France; and 0.79–1.34 µm, Spherotech Inc. Lake Forest, IL, USA). Only events smaller than 0.9 µm were included in the analysis gate. Cell membrane fragments stained using phalloidin-TRITC (Sigma-Aldrich^®^, Merck KGaA, Darmstadt, Germany) at 4 °C for 30 min, in the dark, were excluded from the analysis gate for detection of MSC-EVs. EVs were stained using the appropriate saturating concentrations of conjugated monoclonal antibodies anti-CD9-FITC and anti-CD49-A647 in filtered PBS for 20 min at 4 °C. Staining specificity was further confirmed by subjecting stained EVs following acquisition to 0.05% (*w*/*v*) TX-100 solubilization and reacquiring the sample afterwards. All gated regions were defined using the appropriate fluorescence minus-one (FMO) controls. Fluidic stability, laser alignment, and time delay were checked daily to minimize fluctuations in the fluorescent signal recovered. FACS Canto II analyzer (BD Biosciences, San Jose, CA, USA) equipped with 3 lasers able to discriminate up to 10 fluorophores was used for sample acquisition. Instrument performance was checked daily using CS&T Beads (BD Bioscience, San Jose, CA, USA) and SPHERO Rainbow beads (Spherotech, Lake Forest, IL, USA). Prior to the acquisition of the EV sample, the instrument was cleaned by washing more than 10 times with Clean FACS buffer (BD Biosciences, San Jose, CA, USA) and water in turn, for 10 min each at a high flow rate. The instrument data acquisition and analysis were performed with FACSDiva v.6.2 (BD Pharmingen) and Flow-Jo v.9.7 (Tree Star Inc., Ashland, OR, USA), respectively. 

### 2.8. Cryo-Electron Microscopy of BM-MSC EVs 

Right after preparation of EVs by centrifugation, a 3.5-μL droplet of EVs freshly resuspended in PBS at a final concentration of 0.2 × 10^9^ particles/µL was applied on a copper 300-mesh Quantifoil R2/1 holey carbon grid, previously glow discharged for 30 seconds at 30 mA using a GloQube system (Quorum Technologies, Laughton, UK). After an incubation of 60 seconds, the grid was plunge-frozen in liquid ethane using a Vitrobot Mk IV (Thermo Fisher Scientific, Waltham, MA, USA) operating at 4 °C and 100% RH. Images of the vitrified specimens were acquired using a Talos Arctica transmission electron microscope operating at 200 kV and equipped with a CETA 16M (all from Thermo Fisher Scientific, Waltham, MA, USA) camera. Images with applied defocus values between 3 and 5 μm were acquired with a total exposure time of 2 seconds and a total accumulated dose of 95 electrons per A2 at nominal magnifications of 22,000× or 45,000×, corresponding to a pixel size of 4.7 Å/pixel and 2.3 Å/pixel at the specimen level, respectively.

### 2.9. Western Blotting of BM-MSC Cell and EV Lysates

Western blots of BM-MSC lysates (lysis buffer: 10mM HEPES-NaOH pH 7.5; 2mM EDTA pH8.0; 1% (*w*/*v*) SDS; supplemented with protease and phosphatase inhibitors) at different passages and of EVs at passage 14 were prepared and after addition of Laemmli loading buffer were denatured (10 min at 65 °C) and loaded onto 4–15% Criterion™ TGX Stain-Free™ (Bio-Rad, Hercules, CA, USA) for SDS-PAGE. The proteins were transferred to nitrocellulose membranes using the Trans Blot^®^ Turbo System™ (Bio-Rad, Hercules, CA, USA). 

Membranes were cut according to molecular weight standards in order to maximize incubation with different antibodies and were allowed to react with antibodies directed against the following proteins: Histone H2A.X phospho S139 (1:50000; Abcam, Cambridge, UK), p16^INK4a^ (1:1000; Santa Cruz, Dallas, TX, USA), Neprilysin (1:1000; Merck-Millipore, Burlington, MA, USA), AGO-2 (1:1000; Sigma-Aldrich^®^, Merck KGaA, Darmstadt, Germany), Alix (1:1000; Cell Signaling, Danvers MA, USA), CD9 (1:500; Sigma-Aldrich^®^, Merck KGaA, Darmstadt, Germany), TSG101 (1:1000; Sigma-Aldrich^®^, Merck KGaA, Darmstadt, Germany), CD63 (1:500; Sigma-Aldrich^®^, Merck KGaA, Darmstadt, Germany & Santa Cruz, Dallas, TX, USA) and HSP70 (1:5000; Sigma-Aldrich®, Merck KGaA, Darmstadt, Germany), in 5% (*w*/*v*) BSA (Sigma-Aldrich®, Merck KGaA, Darmstadt, Germany) or skimmed milk (Régilait, Saint-Martine-Belle-Roche, France) in TBS-T (137 mM NaCl, 20 mM Tris-HCl pH 7.5., 0.1% (*v*/*v*) Tween-20). Bands of interest were revealed following incubation with goat anti-rabbit, anti-mouse and anti-Armenian hamster secondary antibodies conjugated with HRP and revealed by SuperSignal™ West Pico PLUS or Femto (Thermo Scientific™, Thermo Fisher Scientific, Waltham, MA, USA). Acquisition of chemiluminescence signals was performed with a ChemiDoc MP system (Bio-Rad, Hercules, CA, USA). Normalization of Western blot signals was performed on the total protein loaded on each lane by the Stain-Free™ technology using the Image Lab™ v.6.0 (Bio-Rad, Hercules, CA, USA) software as previously described [27].

### 2.10. Quantitative RT-PCR 

BM-MSC P14 and primary murine Fibroblasts, prepared from the skin of 3-day-old pups according to previously described protocols [28,29], were solubilized in 500 μL of TRI Reagent^®^ solution (ZYMO RESEARCH, Irvine, CA, USA) for RNA extraction. Total RNA was isolated using the RNA Direct-zol™ MiniPrep Isolation Kit (ZYMO RESEARCH, Irvine, CA, USA) according to the manufacturer’s protocol. After elution in 25 μL DNase/RNAse-free water, the total RNA was quantified using NANOdrop 2000c spectrophotometer (Thermo Fisher Scientific, Waltham, MA, USA) and its quality checked for by 260/280 nm optical density ratio. 1 μg of RNA underwent reverse transcription with the High-Capacity cDNA Reverse Transcription kit (Applied Biosystems™, Thermo Fisher Scientific, Waltham, MA, USA). Quantitative real-time polymerase chain reaction (qRT-PCR) was performed in a final volume of 10 μL by means of a Sybr Green detection kit (SensiFAST™ SYBR^®^ Lo-ROX, Bioline, Paris, France) in a Viia7 Real-Time PCR System (Applied Biosystems™, Thermo Fisher Scientific, Waltham, MA, USA). Each transcript was evaluated with at least duplicate measurements and Neprilysin relative expression was calculated following normalization against that of β-actin with the comparative ΔΔCT method using the values obtained from skin fibroblasts mRNA as a reference. The following primers were used: Neprilysin Forward: 5’-CCCAGTGTATGGTATACCAG-3’; Reverse: 5’-TGGCCAATACCTCCATTATCA-3’; β-actin Forward: 5’-GCCATCCTGCGTTCTGGA-3’, Reverse: 5’-GCTCTTCTCCAGGGAGGA-3’.

### 2.11. Animal Studies

Procedures involving animal handling and care were performed in accordance with protocols approved by the Humanitas Clinical and Research Center (Rozzano, Milan, Italy) in compliance with national (4D.L. N.116, G.U., suppl. 40, 18-2-1992) and international law and policies (EEC Council Directive 2010/63/EU, OJ L 276/33, 22-09-2010; National Institutes of Health Guide for the Care and Use of Laboratory Animals, US National Research Council, 2011). All efforts were made to minimize the number of mice used and their suffering. For in vivo experiments, 5–6 animals per experimental condition were used of the double transgenic Alzheimer’s Disease mouse model APP/PS1 (APPswe-PS1dE9) purchased from Jackson Laboratory [30] (Bar Harbor, ME, USA). Animals were housed and bred in the SPF animal facility of Humanitas Clinical and Research Center. BM-MSC-EVs were resuspended in PBS before intracranial injection into male APP/PS1 mice. The head skin of anesthetized animals (100 mg/kg ketamine/10 mg/kg xylazine) was incised with a scalpel and the skull exposed. The skull surface and the bregma served as the stereotaxic zero points. The cranium was perforated at two sites (AP 1 mm, ML 2 mm, DV 2 mm), once in each hemisphere, and using a Hamilton precision syringe (26G needle), 4 µL of BM-MSC-derived EV suspension (5.6 µg/µL, corresponding to 22.4 μg of MSC-EVs obtained from ~3x10^6^ cells corresponding ~1 × 10^9^ particles) or 4 µL of PBS was injected into the neocortex as previously described [31]. Twenty-five days after treatment, animals were sacrificed according to the approved animal protocol guidelines and intracardially perfused with 4% (*w*/*v*) PFA.

### 2.12. Immuno-Histochemical and Immunofluorescence Staining 

Brains of perfused animals, after further ON fixation in 4% (*w*/*v*) PFA, were washed with PBS and freshly cut by vibratome (Leica VT1000-S vibroslicer). 30 µm thick para-sagittal sections were stored in anti-freezing solution (*v*/*v*: 3% glycerol, 3% Ethylene glycol, 40 mM phosphate buffer, pH 7.4) at −20 °C, following cryopreservation in 7% (*w*/*v*) and 20% (*w*/*v*) sucrose solutions (in 0.1M Phosphate Buffer, pH: 7.4), until use. For immunohistochemical staining, free-floating slices were permeabilized in PBS solution (PBS, 3% (*v*/*v*) Methanol, 0.1% (*w*/*v*) Triton X-100) and treated with 3% (*v*/*v*) Hydrogen Peroxide. For antigen retrieval, slices were treated with 90% (*v*/*v*) formic acid in water. Slices were subsequently washed with 0.1% (*w*/*v*) Triton X-100 in PBS and incubated for 1 h at RT in blocking solution (10% (*v*/*v*) horse serum, 0.1% (*w*/*v*) Triton-X100 in PBS). All slices were stained O/N at 4 °C with 6E10 antibody (β-amyloid, 1-16 Monoclonal Antibody, Covance®, BioLegend^®^, San Diego, CA, USA) in blocking solution. The immuno-detection was performed with an anti-mouse secondary antibody followed by MAC1 Mouse Probe and MAC1 Universal HRP-Polymer (Biocare Medical, Pacheco, CA, USA) and DAB (3’DiAminoBenzidine; Biocare Medical, Pacheco, CA, USA) revelation. Slices were mounted on glass slides with FluorSave™ Reagent (Calbiochem^®^, Merck KGaA, Darmstadt, Germany). 

For immunofluorescence staining, brain sections were incubated for 10 min in 0.04% (*w*/*v*) ThT (Sigma-Aldrich®, Merck KGaA, Darmstadt, Germany) solution, then washed for 1 minute with 80% (*v*/*v*) methanol in water and stained O/N at 4 °C with Iba1 (rabbit anti-mouse, FUJIFILM Wako Chemicals Europe GmbH, Neuss, Germany) and Smi31 and Smi32 monoclonal anti-mouse antibodies (BioLegend®, San Diego, CA, USA). After washing, the secondary antibodies (anti-rabbit secondary antibody conjugated with Alexa 488 and anti-mouse secondary antibody conjugated with Alexa 633 fluorochromes) were incubated for 1h at RT in blocking solution. Slices were mounted on glass slides and mounted with FluorSave™ Reagent (Calbiochem^®^, Merck KGaA, Darmstadt, Germany).

### 2.13. Image Acquisitions and Analysis

Immuno-histochemically stained slices were acquired by Olympus DotSlide microscope with a 10× objective, connected to a computer equipped with OlyVIA^®^ 2.9 viewer software (Olympus, Hamburg, Germany). Quantification of β-amyloid plaques on immuno-histochemically stained brain slices of APP/PS1 mice was performed using ImageJ software. Four parameters were evaluated to describe plaques: Area Mean (µm^2^), Plaque Solidity (a.u.), i.e., β-amyloid intensity within immune-revealed plaques and calculated as [Area]/[Convex area], and Plaque Density (n plaques/ µm^2^ slice area). About 10 slices for each APP/PS1 animal injected with PBS or EVs were analyzed. Within each slice, two areas of interest were manually selected: the cortex and the hippocampus. Within these areas, the total number of plaques was counted. An arbitrary threshold has been fixed and maintained for the whole analysis of slices of the same experimental group. Collected data were averaged and analyzed with Graph Pad Prism6^®^ software. All values were expressed as the mean ± SEM. Graphs represent the average of plaques area (Area, μm^2^), solidity (a.u.) and of plaque density (n/µm^2^) for all analyzed animals (5–6 animals per group). 

Confocal images were acquired using a Leica TCS SP5 confocal microscope equipped with a HCX PL APO lambda blue 63x/1.4 OIL objective. ThT signal was revealed by exciting with 405 nm, Iba1 with 488 nm and Smi31-32 with 633 nm lasers. Voxel size was established using Nyquist criteria. Confocal images were analyzed by IMARIS v7.4.2 software (Oxford Instruments, Abingdon-on-Thames, UK). Briefly, 3D image reconstruction of Z-planes of ThT and Smi31-32 signals was performed and spots of Smi31-32 signal were quantified using the IMARIS software function “Spot” with the following parameters: detected spot number/estimated diameter = 0.962 μm; Background Subtraction Quality between 40 and 200. Spot detection was monitored and corrected manually, setting 90 µm as the minimum spot to be considered as a bona fide Smi31-32 signal. The ThT signal volume was quantified by the “Surface” function, setting Smooth as 0.192, Absolute Intensity, Threshold between 140 and 75. Smi31-32 ROI volume analysis was drawn according to the Iba1 (to detect microglia cells around the plaques) fluorescence around the ThT signal. Smi31-32 spot density was estimated with the following formula: Ratio of [Smi31-32 spots / (Iba1 ROI volume minus ThT volume)]. 

### 2.14. Statistical Analysis 

Average values obtained from plaques quantifications of EVs treated versus non-treated APP/PS1 animals were compared using the unpaired non-parametric Mann-Whitney test, with Prism6^®^ software. In general, the one tail hypothesis was chosen (reduction of all parameters tested) with Welch’s correction. All data are expressed as means ± SEM. *p*-Values lower than 0.05 were considered statistically significant. In the graphs, one * corresponds to *p* < 0.05, two ** corresponds to *p* < 0.01.

## 3. Results

EVs derived from mouse BM-MSCs were investigated for a possible therapeutic effect in APP/PS1 mice. To this end, MSCs were prepared from adult bone marrow and characterized for typical markers, following criteria suggested by the International Society for Cellular Therapy [32], including evaluating their ability to differentiate towards osteogenic and adipogenic lineages.

The flow cytometry characterization shows that BM-MSCs express stemness markers, such as SCA1, along with CD73 and CD105, specific markers for the mesenchymal origin of cells. BM-MSCs were also positive for markers expected to be present on the surface of EVs, such as CD44, CD29, CD49 and CD9 [11]. Lineage Hematopoietic Markers (LIN), CD117 and CD31, were not detected, excluding the contamination of cell cultures by hematopoietic cells or their precursors (Figure 1a). Reliability of BM-MSC preparations was confirmed by the repeatability of the staining for the different markers in 6 independent preparations (Figure 1b). Differentiation of BM-MSCs towards osteogenic (Alizarin Red staining) and adipogenic (Oil-red O staining) lineages further confirms the identity of our cell cultures (Figure 1c) [33]. Passages from 9 to 14 have been used since mouse BM-MSCs, differently from other species (i.e., human and rat), maintain a high contamination of hematopoietic cells for a longer time in culture [34]. Possible senescence was evaluated by comparing SA-β-Gal activity in BM-MSC at P6 and P14, representative of early and late passages of the cultures used in our experiments (Figure 1d). The percentage of β-Gal positive cells was similar in the two passages (Figure 1e). Furthermore, two other markers of senescence, the phosphorylation of Ser-139 of histone H2A.X (Figure 1f), indicative of oxidative stress, and the protein level of the cell cycle regulator p16^INK4A^ (Figure 1g), were investigated by Western blot. Lysates from BM-MSCs at passages ranging from P5 to P17 were compared with cells treated with H_2_O_2_ to induce oxidative stress (positive control), showing that no significant difference in senescence processes activation was detected in the passages of BM-MSCs used for our experiments compared to earlier ones. 

EVs secreted by BM-MSCs into the cell culture medium were purified by ultracentrifugation following a widely used protocol yielding a pool of microvesicles and exosomes [26]. Nanoparticle tracking analysis (NTA) allowed the determination of the size and the concentration of isolated EVs (Figure 2a). The results of 5 independent preparations show the three most represented vesicle populations, associated with three peak sizes (two >100 nm, one ~400 nm), indicating the successful isolation of a pool of EVs, including both exosomes and microvesicles. The mean concentration of the major vesicle peak (130 nm) was 1.48 × 10^6^ particles/mL (± 88027 SEM) (Figure 2a, left panel). The total concentration measured as the area under the curve (AUC) of the mean values of all preparations tested resulted in 1.4 × 10^9^ particles/mL (Figure 2a, right panel). The analysis of EV-depleted supernatants showed a mean particle concentration of 3.18 × 10^5^ particles/mL (± 26123 SEM) and a major peak at 94.50 nm (Figure 2a, green line), indicating an efficient yield of the procedure for EVs larger than 100 nm. The analysis by cryo-electron microscopy performed on EV preparations just after centrifugation showed the presence of vesicles of different sizes compatible with exosomes and microvesicles, surrounded by a lipid bilayer and with a variably dense content, confirming furthermore the integrity and the identity of the vesicles (Figure 2b). A first characterization of EVs’ specific surface markers was performed by flow cytometry. In particular, CD9 and CD49, even though the latter was expressed at a lower extent, were detected on BM-MSC-EVs (Figure 2c). The gating strategy used beads allowing selection of events below 1.34 µm. Using Phalloidin to label cellular debris, we further gated against contaminating events that could fall within this size. To further ensure that the analyzed events were within the size range of EVs, we also performed a calibration with Megamix beads (Biocytex) ranging from 0.1 µm to 0.9 µm using the same acquisition parameters used for EV analysis, showing that most of the gated events positive for CD9 and CD49 were in the range between 0.1 and 0.9 µm (Appendix A). To support the vesicular origin of CD9 and CD49 events, antibody-stained EVs, following first flow cytometry acquisition, were treated with 0.05% (*w*/*v*) Triton-X100 for 30 min and re-acquired afterwards for the same time, showing a massive reduction of (CD9^+^/CD49^+^) events thereafter (Figure 2d). This supports the vesicular nature of these events [35]. In addition, well-established EV markers, such as Alix, CD9, CD63, HSP70, AGO2 and TSG101, were identified by Western blots in P14 EV lysates obtained from approximately 1 × 10^9^ particles and compared to BM-MSC lysates (Figure 2e). 

Since previous work reported the presence of Neprilysin (NEP), an enzyme able to exert Aβ-degrading activity, on human ADSC and their exosomes, we also tested NEP expression in BM-MSCs and their derived EVs. Interestingly, NEP could be detected in the EVs’ lysates and its mRNA was also expressed in P16 BM-MSCs, at a level more than 100 times that of murine fibroblasts (Figure 2f).

The earliest signs of cognitive impairments in APP/PS1 mice have been reported in six-month-old animals [36]. Conversely, amyloid plaque deposition has been shown to start at approximately six weeks of age in the neocortex and at about three to four months of age in the hippocampus [37,38]. To assess the effects of BM-MSC-EVs in the APP/PS1 AD mice at early stages of the disease, before the behavioral manifestations become apparent, we performed intraparenchymal injections of EVs in the cortex of mice at 3 and 5 months of age and evaluated, 25 days later, EV effects on two of the typical early signs of the disease, Aβ deposition and appearance of dystrophic neurites (Figure 3 and Figure 4). No gross AD-unrelated behavioral alteration was observed in mice treated with EVs compared to PBS-treated ones. Brain sections from hippocampus and cortex of EV-treated APP/PS1 compared to mice injected with vehicle (PBS, referred to as the controls) were immunohistochemically stained with 6E10 antibody, reacting with the N-terminal domain of human Aβ_1-42_ peptide. The injection of BM-MSC-EVs in 5-month-old mice (Figure 3) resulted 1 month later (i.e., at 6 months) in a reduction of Aβ plaque area in both hippocampus and cerebral cortex, depicted by two representative images with their close-up views on plaques (Figure 3a). Moreover, plaque solidity, a parameter that represents Aβ loading of plaques, was strongly reduced, in particular within the neocortex (Figure 3b), the region in which the injection has been performed. In addition, the plaque density (number of plaques/μm^2^) was significantly affected by EV treatment in the hippocampal region, but not in the cortex (Figure 3b). 

We then performed the same analysis in 4-month-old mice (Figure 4) displaying significantly smaller and a lower number of plaques relative to the 6-month-old mice (compare the area and density in the control animals in Figure 3b and Figure 4b). The injection of BM-MSC-EVs was able to decrease the average plaque area, mostly in the hippocampus, and reduced the density (number of plaques/μm^2^) both in the hippocampus and in the cortex when compared to the age-matched controls (Figure 4b). On the basis of these results, we can hypothesize that EVs may operate not only by promoting the disaggregation of Aβ pre-existing deposits (Figure 3), but also by preventing or slowing down the formation of new plaques (Figure 4). 

It is known that, in neurons, microtubule disruption and microtubule-based transport impairment, as well as neurofilament disorganization, lead to dystrophic neurite formation [39,40]. Along with dysfunction associated with axonal swelling and impaired transport, the axonal dystrophy is likely to exacerbate downstream neurodegeneration, leading to cognitive deficits. In an attempt to investigate whether the reduction of Aβ plaques resulted in amelioration of neurite morphology, we analyzed dystrophic neurites in the cortex and hippocampus of AD mice. Immunostaining of brain sections 25 days following injection of EVs at both ages was performed with Smi31-32 antibodies, which recognize neurofilament H and M [41] (Figure 5). ThT-positive plaques were surrounded by dystrophic neurites (Smi31-32-positive) at both ages analyzed, with the results being significantly higher at 6 months than at 4 months (Figure 5a). Notably, the injection of BM-MSC-EVs reduced the number of Smi31-32 positive dots in the area of the plaques. Although the decrease was evident at both ages, the results were only statistically significant at 6 months (Figure 5b,c). 

## 4. Discussion 

MSC-derived EVs are increasingly attracting the attention of the scientific community as possible therapeutic tools. By promoting neurogenesis, angiogenesis and remodeling of nervous processes and being endowed with immunomodulatory actions [42,43,44,45], MSC-EVs might exert protective roles in the brain. 

In the last few years, the possibility of exploiting the therapeutic potential of EVs has been evaluated in AD preclinical models, where encouraging outcomes have been achieved [11,46,47]. 

The studies reported so far, however, have explored the therapeutic potential of MSC-EVs at a stage in which the pathology is already developed and the cognitive deficits are overt, i.e., a condition that is difficult to revert. In particular, two different labs have investigated the effects of EVs on AD mice when the pathology is clearly manifested: Cui and collaborators have shown that 7-month-old APP/PS1 mice, treated twice a week for 4 months with exosomes extracted from hypoxic MSCs, improved cognitive impairments compared to untreated AD mice [47]; Wang and coworkers conclude from their study that MSC-derived EV treatment of AD mice suppress iNOS expression and ameliorates cognitive behavior by partially rescuing the Aβ-induced deficits in hippocampal synaptic plasticity [48]. We therefore wondered whether carrying out the intervention earlier could lead to positive outcomes, possibly slowing down AD progression. Even if it is still probably too early to answer this question, our results could represent a perspective in this direction. In the present study, we assessed the effects of BM-MSC-EVs in AD mice at a young age when the plaques are just beginning to appear and the overt cognitive deficits are not yet manifested.

Following extensive characterization of EVs obtained from murine BM-MSCs, we showed that they are a mixed population mostly composed of exosomes and microvesicles, endowed with well-established markers and preserving their integrity following purification, as shown by Cryo-EM. To investigate the actions induced by BM-MSC-derived EV injection, we compared APP/PS1 mice of 3 and 5 months of age bilaterally injected into the cortex with vehicle (PBS, controls) or MSC-EVs. Our data suggest that EVs may operate not only by promoting the disaggregation of Aβ pre-existing deposits (5 months => 6 months), but also preventing or slowing down the formation of new ones (3 months => 4 months). This in turn could account for neuroprotective effects, as suggested by the reduction of dystrophic neurites in treated mice. 

What are the mechanisms involved in MSC-EV effects? We can speculate that on the one hand, EVs might act directly on the plaques, inducing their disaggregation through an interaction between EV lipid membranes and Aβ plaques. EVs have been shown to also act as amyloid scavengers based on Aβ_1-42_ ability to bind to glycosphingolipids that are extremely abundant in exosomes [20]. This would lead to phagocytosis of Aβ -along with EVs binding to it- by phagocytic cells (i.e., microglial cells), in line with the hypothesis that MSC-EVs could promote Aβ clearance by a direct binding. This possibility would be consistent with the decrease in the plaque area, as well as in the amount of Aβ within plaques (solidity) observed in treated mice. 

On the other hand, MSC-EVs have been described to carry the enzymatically active NEP, a type II membrane-associated metalloendopeptidase involved in the proteolysis of Aβ [19,49]. When added to N2a line cells, AD-MSC exosomes reduced both extracellular and intracellular Aβ_1-42_ deposits [19]. Along this line, the presence of NEP in the lysates of both mouse BM-MSC and their derived EVs, together with the detection of mRNA expression in the cells (surprisingly high when compared to fibroblasts) suggests that this could be a possible mechanism explaining EV action on Aβ-plaques in 3- and 5-month-old APP/PS1 mice. In addition, EVs might also act on microglia cells. In fact, in AD brains microglia clusters around Aβ_1-42_ deposit and acquire a polarized phenotype with hypertrophic processes extending towards plaques [50,51,52]. Microglia are thought to regulate the degree of amyloid deposition by phagocytosis of amyloid aggregates with a potentially protective impact on AD progression [53,54]. Therefore, EVs could act by enhancing microglia functionality and phagocytosis/degradation ability [51].

Furthermore, MSC-EVs have been reported to induce antioxidant effects. De Godoy and colleagues proposed that MSC-EVs, thanks to their content including antioxidant enzymes and anti-inflammatory and/or trophic molecules, exert a neuroprotective action. The authors demonstrated that EVs secreted by MSC contain and carry catalase that endows EVs of reactive oxygen species (ROS) scavenging activity [55].

It is likely that MSC-EVs operate through pleiotropic mechanisms, involving a combination of the above processes, including the inhibition of inflammatory responses, regulation of the immune system, as well as by carrying biologically active molecules, which in turn may act either directly on amyloid plaques or on microglial cells, which are the main responsible for Aβ phagocytosis in the brain.

To clarify EV mechanisms of action, a deeper characterization of the content of EVs and of the different types that participate in tissue repair [56,57] is mandatory. To this end, also exploring novel purification protocols able to recover also the smallest fractions of EVs is urgently needed [58]. It has been reported that EVs contain, in addition to the above-mentioned lipids, proteins and nucleic acids (miRNA, mRNA). Among these, miRNAs inside the lumen could be horizontally transferred to target cells, contributing to the gene expression regulation. For instance, miR-21 overexpression through engineered EVs has been shown not only to decrease plaque deposition but also to down-regulate the levels of TNF-α and IL-1β therefore inducing anti-inflammatory effects [47].

In this study, we focused our attention on EV actions on Aβ plaques and dystrophic neurites, which represent one of the typical hallmarks of AD. We are aware that the correlation between plaque load and cognitive scores in humans has recently been questioned [59,60], prompting the scientific community to look for additional factors, beyond the “Amyloid Hypothesis”, as the etiological mechanism of AD. However, our results represent a clear indication that plaque formation could be delayed by BM-MSC-EVs and that this may significantly reduce the extent of dystrophic neurites. Dystrophic neurites forming around plaques have been recently found to evolve toward progressively more degenerated forms [61], and to promote the accumulation of the amyloid precursor protein (APP) cleaving enzyme (BACE1), which is required for Aβ generation [39]. Given all these processes lead to exacerbation of amyloid pathology in AD, the MSC-EVs-mediated reduction of dystrophic neurite formation may represent an important cue for slowing disease progression.

Moreover, since MSCs have a high immunoregulatory potential and their derived EVs maintain such ability, next studies will be focused on the analysis of the anti-inflammatory action and how it could cooperate with Aβ degrading action in AD therapy. Indeed, MSC-EVs can induce anti-inflammatory effects through the regulation of cytokine release [47,55,62] or iNOS inhibition or by preventing Aβ oligomer toxicity [48], given that iNOS is a common target of many inflammatory pathways and an important endpoint for the therapeutic effects of EVs.

Finally, we believe that EVs and their abilities inherited from BM-MSCs could represent an applicable, safe and cost-effective approach in cell-free regenerative medicine to improve therapy for preventing the earliest stages of the disease to proceed towards AD. We can venture the hypothesis that our data could support the use of EVs in patients when the first signs of Mild Cognitive Impairment begin to appear. Nonetheless, before translating their use into the clinic, a better understanding of EV actions and their biological functions need to be undertaken, together with the choice of the best route of administration. In this respect, intranasal administration, being less invasive, easy and cheap could represent a valid option. 

In conclusion, our results highlight that intracortical delivery of bona fide murine BM-MSC-EVs to APP/PS1 mice at 3 months, before overt clinical signs, is able to prevent Aβ plaque formation and in 5-month-old mice can reduce dystrophic neurons occurrence. As far as we know, this study represents the first evidence of a possible preventive effect of BM-MSC-EVs in an animal model of AD. This result can be relevant if we consider that until now clinical trials and treatments adopted for AD therapy did not produce the desired results, probably because started too late. 

## Figures and Tables

**Figure 1 cells-08-01059-f001:**
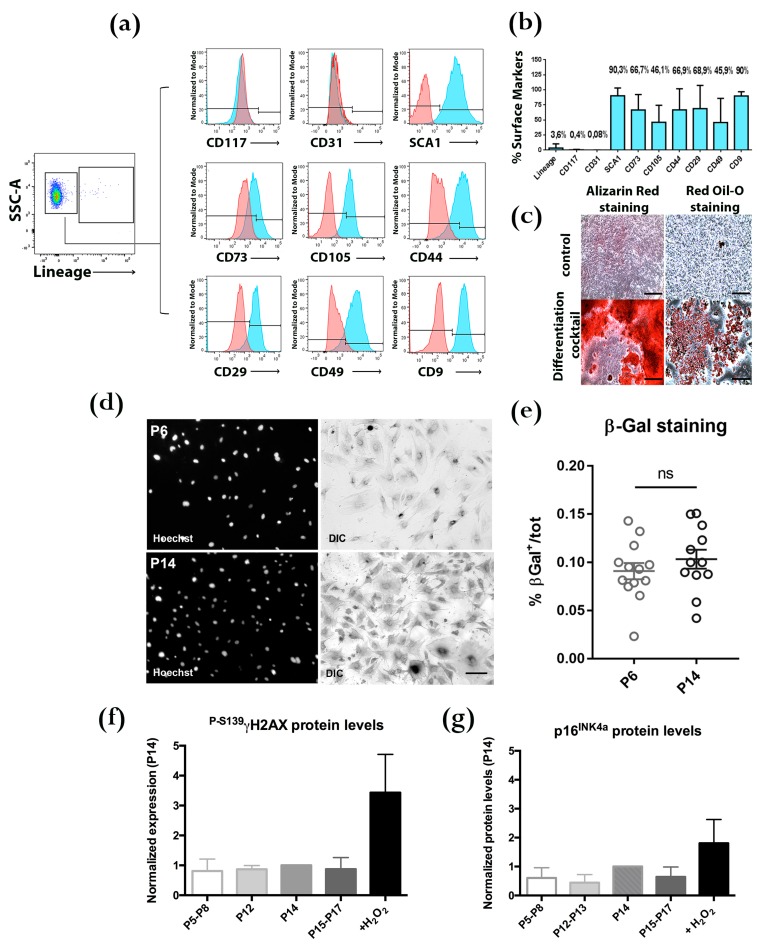
Characterization of murine Bone Marrow Mesenchymal Stem Cells (BM-MSCs). (**a**) Representative panels of flow cytometry analysis showing surface markers of BM-MSCs between passages 6 and 14. Lineage Hematopoietic Markers (LINEAGE) negative cells (96.4 ± 8.4%) are also negative for CD117 and CD31, excluding contamination by hematopoietic cells; they are strongly positive for SCA1 (stemness marker) for CD73 and CD105 (mesenchymal markers) and for CD44, CD29, CD49 and CD9 (markers also found on EVs). (**b**) Percentages of surface marker positive populations (mean ± SEM; flow cytometry experiments of 6 independent BM-MSC cultures). (**c**) Representative images showing differentiation of BM-MSCs into OSTEOCYTES (Alizarin Red staining) and ADIPOCYTES (Oil-red O staining) lineages. Controls: untreated BM-MSC cultures (upper row). Scale bars: 400 μm. (**d**) β-Gal assay was performed on BM-MSCs at P6 and P14 to test senescence of cells maintained in culture. Cells positive for X-gal were quantified and compared to the total population; no significant differences between the passages were detected (*p* = 0.3462; statistical analysis was performed by ANOVA test). Scale bar: 100 µm. (**e**) Representative images for X-Gal of BM-MSCs at P6 and P14. (**f**,**g**) Histone H2A.X phospho S139 (**f**) and p16^INK4A^ (**g**) Western blot analysis, showing that senescence processes are not differently activated during cell passages in vitro.

**Figure 2 cells-08-01059-f002:**
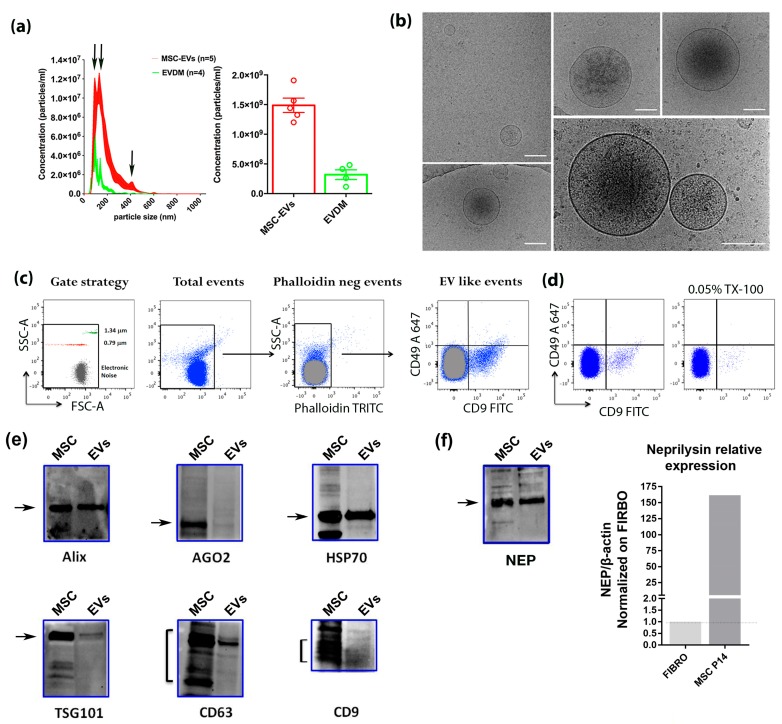
Characterization of BM-MSC-derived EVs. (**a**) Left panel: Nanoparticle Tracking Analysis (NanoSight NS300) of BM-MSC-EVs isolated by ultracentrifugation (red) compared to EV-depleted medium (EVDM) (green). Particle size distribution (left panel) shows that three populations corresponding to ~100 nm, ~150 nm, ~400 nm are the most represented on the red plot (arrows), indicating isolation of different pools of vesicles (> 100 nm exosomes and microvesicles). The green plot shows that fewer and smaller-sized vesicles (average size = 94.50 nm) remain in the medium after ultracentrifugation. Right panel: total particle concentration of isolated BM-MSC-EVs (red histogram) compared to the EVDM (green histogram). All data represent means and standard error of the mean (SEM) of 5 (BM-MSC-EV) and 4 (EVDM) independent preparations. (**b**) Visualization of purified BM-MSC-EVs by cryo-EM: a heterogeneous population of EVs surrounded by a lipid bilayer of sizes compatible with exosomes and microvesicles with an electron-dense core was observed. Note the integrity of membrane vesicles. Scale bar = 150 nm. (**c**) Representative panels showing flow cytometry analysis of EVs from BM-MSCs. Arrows between panels indicate the consecutive gating strategies for the analysis. The “gate strategy” panel shows the forward (FSC-A) and side (SSC-A) scatter density profile of 0.79 μm (red dots) and 1.34 μm (green dots) dimensional beads, which were used to include all events (< 1.34 µm) for specific marker analysis. Smaller events defined as “Electronic Noise”, falling below the limit of resolution of the instrument were also included in the “Total events” gate. BM-MSC-EVs appear in the “Total events” panel. Events falling within the “Electronic Noise”, represented in gray, were subtracted from all analysis gates. Bona fide EVs were Phalloidin negative (“Phalloidin neg events”) and were shown to be mainly CD9-FITC+, CD49-A647^+^ or double positive (“EV-like events”). (**d**) Adding 0.05% (*w*/*v*) TX-100 for 30 min after acquisition (left panel) resulted in a major decrease in the CD49^+^/CD9+ stained population (right panel). (**e**) EVs and BM-MSC lysates (5µg) were analyzed by immunoblotting with antibodies against EV proteins Alix, AGO2, HSP70, TSG101, CD63 and CD9. (**f**) Left panel: Immunoblotting showing the presence of Neprilysin on EVs and in BM-MSC lysates (5 µg). Right panel: Real-time PCR analysis of Nep relative expression in P14 BM-MSC RNA normalized to β-actin and to the expression in murine primary fibroblasts.

**Figure 3 cells-08-01059-f003:**
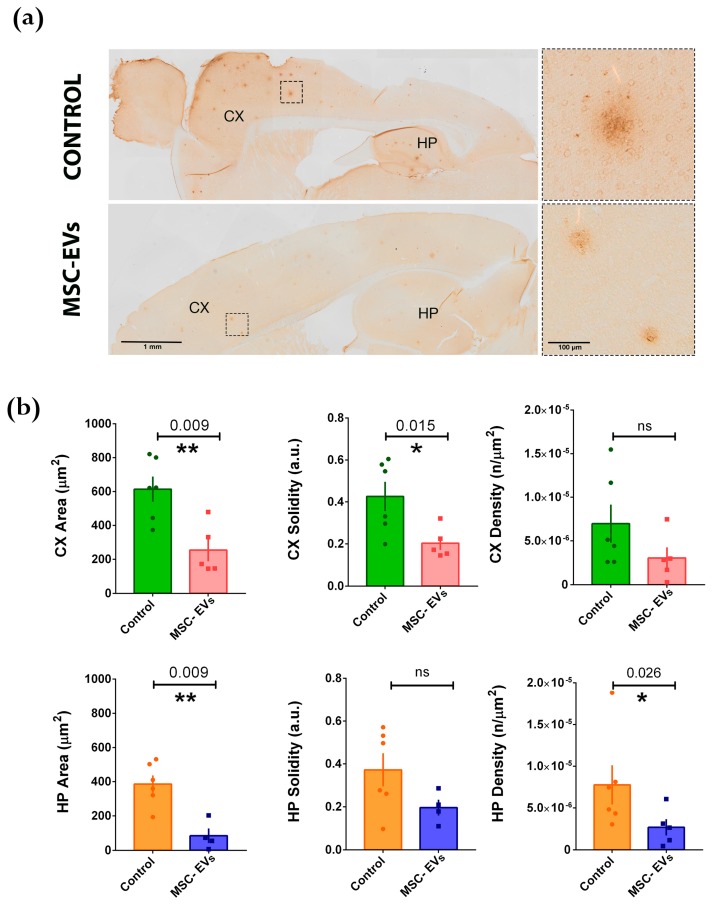
Intracerebral injection of BM-MSC-EVs (MSC-EVs) into 5-month-old APP/PS1 mice reduces amyloid deposition after ~1 month. (**a**) Immunohistochemical staining (DAB) of Aβ_1-42_ plaques in brains of APP/PS1 mice treated with vehicle (CONTROL, top) or MSC-EVs (bottom). Scale bar: 1 mm. Panels on the right show higher magnification of representative plaques, contoured in the inset. Scale bar: 100μm. Cortex (CX) and Hippocampus (HP) are indicated. (**b**) Quantification of Aβ_1-42_ positive plaques in the Cortex (CX; upper graphs: green and red histograms) and in the Hippocampus (HP; lower graphs: orange and blue histograms). Each dot represents an animal for which 10 slices have been scored. Statistical analysis was performed by non-parametric Mann-Withney test (* *p* < 0.05; ** *p* < 0.01).

**Figure 4 cells-08-01059-f004:**
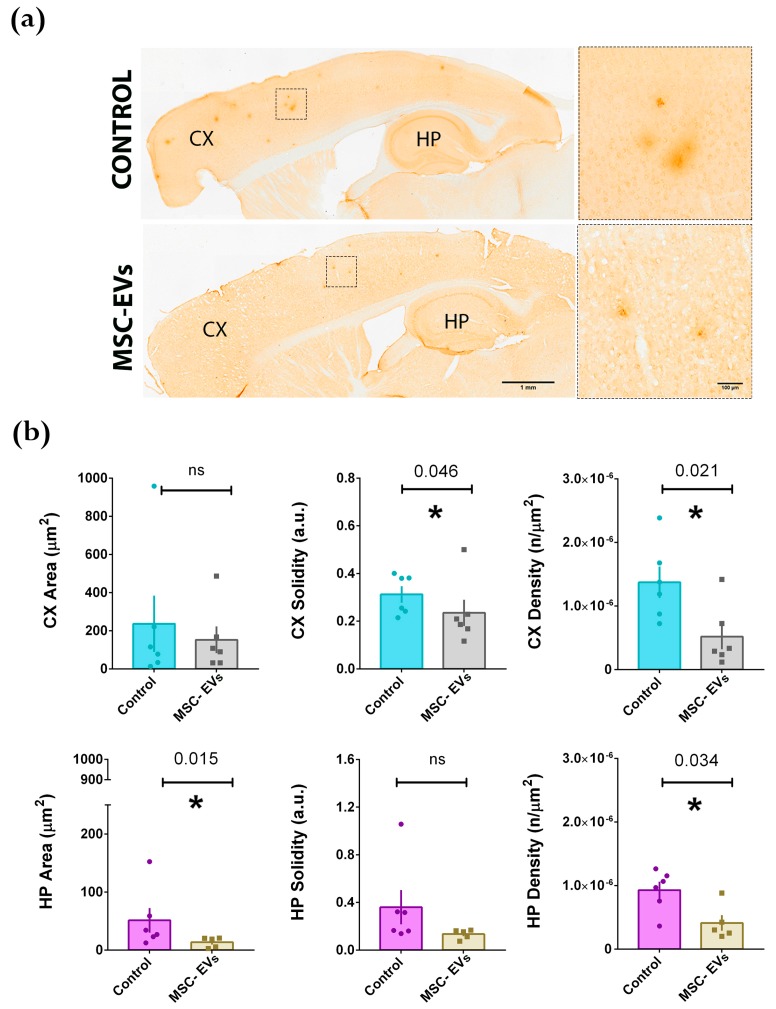
Intracerebral injection of BM-MSC-EVs (MSC-EVs) into 3-month-old APP/PS1 mice reduces amyloid deposition after ~1 month. (**a**) Immunohistochemical staining (DAB) of Aβ1-42 plaques in brains of APP/PS1 mice treated with vehicle (CONTROL, top) or MSC-EVs (bottom). Scale bar: 1 mm. Panels on the right show higher magnification of representative plaques, contoured in the inset. Scale bar: 100 μm. Cortex (CX) and Hippocampus (HP) are indicated. (**b**) Quantification of Aβ1-42 positive plaques in the Cortex (CX; upper graphs: light blue and grey histograms) and in the hippocampus (HP; lower graphs: violet and brown histograms). Each dot represents an animal for which 10 slices have been scored. Statistical analysis was performed by non-parametric Mann-Whitney test (**p* < 0.05; ** *p* < 0.01).

**Figure 5 cells-08-01059-f005:**
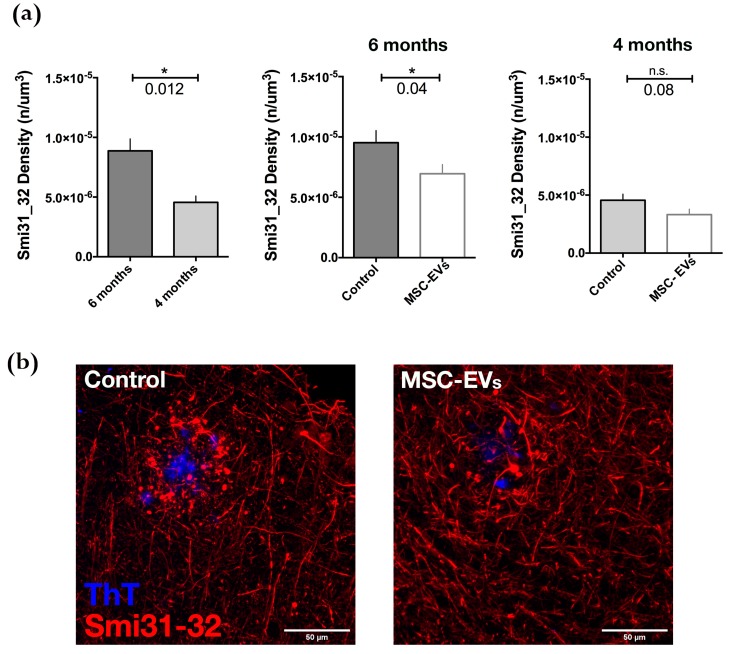
Dysmorphic Neurites around Aβ plaques in APP/PS1 brain cortex slices are reduced following MSC-EVs treatment. (**a**) Left panel: Quantification of Smi31-32 spots (density), representing dystrophic neurites, in the 3D reconstructed ROIs acquired around plaques in brain slices of 6- (dark grey histogram) and 4-month-old (light grey histogram) control APP/PS1 mice. Smi31-32 signal density in 6-month-old APP/PS1 brains is significantly reduced (central panel, white histogram), but not at 4 months (right panel). Statistical analysis was performed by non-parametric Mann-Whitney test (* *p* < 0.05). (**b**) Representative max projections of confocal images of brain slices of 6-month-old APP/PS1 mice 25 days following treatment with EVs (MSC-EVs) or with PBS (Control). Immunostaining with ThT (blue, Aβ plaques) and Smi31-32 (red, dystrophic neurites) are shown. Scale bars: 50 µm.

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
