# Peer review of "Intracerebral Injection of Extracellular Vesicles from Mesenchymal Stem Cells Exerts Reduced Aβ Plaque Burden in Early Stages of a Preclinical Model of Alzheimer’s Disease"

_cells, 2019, doi:10.3390/cells8091059_

Round 1
Reviewer 1 Report
Elia paper described the potential of MSC-derived EVs for AD. The novelty, as the authors stated, is the young age of mice, in a phase where AD is still at the beginning or even without synptoms. The manuscript is clearly written although some major concerns arise:
- Materials and Methods 2.1: Authors state that cells are used at passage 9-14. Why so late passage? Usually, if this research should be translated in clinical practice, MSC are culture up to passage 5 to avoid senescence. Authors should explain the reason of this choice and clearly show in the Results that senescence at passage 9-14 is not different than senescence at earlier passages;
- Result: EV characterization. Flow cytometry of EV is presented. My main concern is that the calibration of the flow cytometer is performed with 0.79 and 1.34 um beads. This means that there is not a real validation of events smaller than at least 0.5 um. Therefore, proposed results are not convincing since all the events shown are a mixture of elecronic noise and possible EVs that fall in a smaller range 0.05-0.5 um. Authors shall present calibration of the instruments with smaller beads (some producers have beads starting from 0.1 um or even smaller). Moreover, FACS Canto has a resolution that is not in the range of small vesicles, since the smallest particles resolved should be no less than 0.2 um. Therefore, how authors can be sure that almost 50% of events are not correctly detected or lost? A clear picture of better characterization of EVs by flow cytometry must be presented, together with wider panel of Ab, such as CD63 and CD81 and negativity for markers of cell contamination. Western Blot should laso be presented. Moreover, electron microscopy must be presented, to confirm that procedure does not crash the EV. Alterantively, CFSE staining of EVs in flow cytometry can be shown, always taking in mind the dimensional issues.
- Results: EVs effect on AD mice. Is it possible to follow the fate of EVs after injection? What about their presence in the days or weeks after administration? And what about their clearance in other body organs?
- Discussion: It is stated that in the literature the presence of Neprilysin is associated with EV. No data about this are presented on the EV used and this should be clearly shown since it could be a crucial mode of action in the AD mice model. So authors have to present data about Neprilysin presence on EVs in the Results.
Author Response
See the pdf enclosed file.

Reviewer 2 Report
Elia et al., showed that exosomes collected from bone marrow mesenchymal stem cells significantly reduce Aβ plaque in brains of Alzheimer's disease model mice. An easy-to-use image digitization method has been selected for the determination of Aβ plaque. The experiments are conducted carefully, and the interpretation of the results is simple. Therefore, I think this paper will promote research on Alzheimer's disease in the future. This paper recommends the correction of the following items and the addition of sentences. (ABSTRACT) Line33:intracerebrally injected Please specify the location of the brain correctly. Line 30:β-amyloid degrading-activitie Please define whether β-amyloid degradation is a direct action of MSC-EV or an indirect action via another type of cell. (Introduction) Line 60:{Cui, 2018) Please add to citations. Line 62: the potential side effects associated with the use of stem cells Please add a concrete example. Line 63:{Liang, 2014}. Please add to citations. Line 63: In in vitro experiments, MSC-derived exosomes significantly decreased both secreted and intracellular Aβ levels in N2a cells engineered to overexpress human Aβ [13]. Please describe this mechanism. If this mechanism is unknown please write unknown. (Experimental conditions without cells are required to prove that MSC-derived exosomes directly degrade Aβ) Line 66:amyloid‐β degrading Please define whether β-amyloid degradation is a direct action of MSC-EV or an indirect action via another type of cell. Line 66: neurotrophic activities Please rewrite to specific words. Line 77: injected into the brains Please specify the location of the brain correctly. (Materials and Methods) Line 88: CO2 Misspelling Line 90:stored at -80°C. Please add the name of the frozen stock solution. Line 91:used from passages 9 to 14 (p9-p14) Please add the scientific basis for using p9-p14 in this experiment. The use of p3-p5 is recommended for adipose-derived mesenchymal stem cells for clinical. Line 108:LSR Fortessa analyzer BD LSRFortessa analyzer Line 115:3x106 Misspelling Line 115:75cm2 Misspelling Line 118:to pellet extracellular vesicles (EVs) Please specify whether the protein fraction contained in the culture supernatant has been removed. Line 168: 4μl BM-MSC-derived EV suspension (5.6 μg/μl) or saline were injected into the neocortex as previously described Please add information on the injection needle. Please describe the reason the dose volume was set to 4 µl (The volume of 4µl is considered to be excessive to the brain of the mouse). Line 199: Line 200: Line 206: Line 206:µm2 Misspelling (Results) Figure 1 The characters are too small to read. Please enlarge the characters of Figure 1(a)-(e). Figure 1(d) Please write the characters of CD9 and CD49 into the figure. Line 241:yielding a pool of microvesicles and exosomes [15]. Please specify whether the protein fraction contained in the culture supernatant has been removed. Line 253:1.48 x 106 Misspelling Line 254:1.4x109 Misspelling Line 256:3.18 x 105 Misspelling Line 301:An unspecific activation of phagocytic microglia activity induced per se by intracerebral injection can be ruled out by the fact that EVs injected animals have a reduced Aβ plaque burden both in the cortex (injection site) and hippocampus (Figure 2b). In this study, the administration solution was administered into the neocortex. However, it can not be denied that the administration solution leaked into the cerebral ventricle. Therefore, comment on activation of microglia in the whole brain is to be made carefully. Figure 2(a), 3(a) Please write the location of the penetration point and the name of the brain part (Cx, HP) in the picture. CTRL is Control. In addition, please describe as Saline when you have administered brine it. Please delete the characters of APP/PS1 and x10. In Figures 2 (a) and 3 (a), unify the left and right direction of the photo. Figure 2(b), 3(b), 4(a), 4(b) Please add a horizontal line below the star that indicates a significant difference. Line 335: 345: .. Misspelling Figure 4(a), 4(b) The color of the bar misleads the reader. Please change the color of the bar. Figure 4(c) It is difficult to see the characters of ThT described in the photo. Please improve it. (Figure legend) Figure4(b) Fig. 4 (b) Please add the explanation of the right graph. Line 319:we analyzed dystrophic neurites in the cortex and hippocampus of AD mice. Please add the information about the sample collection site (cortex or hippocampus) to Figure legend of Figure 4. (Discussion) Please describe the presence or absence of physiological and behavioral changes unrelated to AD in mice that were injected MSC-EV in neocortex. Please indicate if there were any improvement in behavioral testing for AD with APP/PS1 mice that were injected MSC-EV in neocortex. 5.6 ug/μl × 4 μL of MSC-EV was injected in neocortex to one mouse. Please describe the number of MSC cells required to obtain this amount (5.6 x 4 = 22.4 μg) of MSC-EV. Please explain why you did not administer MSC-EV intravenously. Is there a need to further improve this experimental result? Please comment on this treatment of AD for human clinical trials. Line 383:by phagocytic cells Please specify the specific cells name. Line 389: There is no period. Line 422:(Morris Losurdo and Silvia Coco, personal communication) Please specify this URL. (Conclusion) Please rewrite the conclusion. And please describe the most important results of this study.Author Response
See the enclosed pdf file.

Reviewer 3 Report
Reference of your paper, highly related to topic of the manuscript should be included.
Extracellular Vesicles from Mesenchymal Stem Cells Exert Pleiotropic Effects on Amyloid-β, Inflammation, and Regeneration: A Spark of Hope for Alzheimer's Disease from Tiny Structures?
Elia CA, Losurdo M, Malosio ML, Coco S.
Bioessays. 2019 Apr;41(4):e1800199. doi: 10.1002/bies.201800199.
MSC exosomes are highly heterologous population of extracellular vesicles, which is reflected also in their biological activity. Ultracentrifugation does not sediment all biologically active exosomes.
The use intranasal administration of exosomes as an easy, cheap, and safe alternative route for treatment of neural
disorders. It should be discussed.
It would be interesting to test human MSC exosomes in the mice model.
I missed the number of mice used in the study. Is a period of time 25 days after the
animal treatment enough for the conclusion of complete Alzheimer disease prevention?
Author Response
See the enclosed pdf file.
